# Osteoprotective Effect of Pine Pollen in Orchidectomized Rats

**DOI:** 10.3390/nu17132110

**Published:** 2025-06-25

**Authors:** Paweł Polak, Radosław P. Radzki, Marek Bieńko, Sylwia Szymańczyk, Kinga Topolska, Małgorzata Manastyrska-Stolarczyk, Jarosław Szponar

**Affiliations:** 1Department of Orthopedics and Traumatology, Provincial Specialist Hospital in Biała Podlaska, ul. Terebelska 57-65, 21-500 Biała Podlaska, Poland; pawel.polak@szpitalbp.pl; 2Department of Animal Physiology, Faculty of Veterinary Medicine, University of Life Sciences in Lublin, Akademicka 12, 20-033 Lublin, Poland; sylwia.szymanczyk@up.lublin.pl (S.S.); malgorzata.manastyrska-stolarczyk@up.lublin.pl (M.M.-S.); 3Department of Plant Product Technology and Nutrition Hygiene, Faculty of Food Technology, University of Agriculture in Kraków, al. Mickiewicza 21, 31-120 Kraków, Poland; kinga.topolska@urk.edu.pl; 4Toxicology Clinic, Clinical Department of Toxicology and Cardiology, Stefan Wyszyński Regional Specialist Hospital, Medical University of Lublin, 20-718 Lublin, Poland; jaroslaw.szponar@umlub.pl

**Keywords:** bone metabolism, bone mechanics, orchidectomy, male osteoporosis, pine pollen, densitometry, peripheral quantitative tomography

## Abstract

**Background/Objectives**: This study aimed to establish the potential osteotropic effect of pine pollen on bone metabolism in male rats during the development of osteopenia induced by orchidectomy (ORX). We also established the effect of gonadectomy and pine pollen on the characteristics of calf muscles. **Methods**: This study was conducted using 40 male Wistar rats divided into one sham-operated (SHO) and four ORX groups. The SHO rats and one ORX group (negative control) were treated with physiological saline (PhS). The remaining ORX groups received exclusively testosterone (positive control) and two doses of pine pollen (50 and 150 mg/kg b.w.), respectively. The rats were killed 60 days later and their right tibia and left pelvic limbs were isolated. The tibia was analyzed using densitometry, computed tomography, and a bending machine to determine densitometry, structure, and mechanical properties, respectively. The left pelvic limb allowed for measurements of area, density, and fat tissue in the calf muscle. **Results**: The dose of 150 mg/kg b.w. inhibited the development of atrophic changes, both in the cortical and trabecular bone tissue. The dose of 50 mg/kg b.w. also has a protective effect on bones but is less pronounced and concerns only the trabecular bone tissue. The higher dose of pine pollen inhibited the catabolism of the calf muscles by maintaining the density and surface area as in the SHO group. It also limited the accumulation of intramuscular and subcutaneous adipose tissue. **Conclusions**: It is worth emphasizing the osteoprotective effectiveness of pine pollen, especially when administered in larger doses, which demonstrates the possibility of its use in the prevention of the development of osteoporosis in males.

## 1. Introduction

The World Health Organization defines osteoporosis as a systemic skeletal disease characterized by low bone mass, impaired microarchitecture of the tissue, and increased susceptibility of bones to fractures. For a long time, osteoporosis was considered a major health problem affecting mostly women. However, osteoporotic fractures also occur in men. Moreover, the increase in the incidence of osteoporosis and fractures in recent decades has been more pronounced in men than in women. The chance that males will suffer an osteoporosis-related fracture during their lifetimes is greater than the chance of developing prostate cancer [1]. In contrast to women, where menopause is associated with a rather rapid decline in estrogen levels, in men, androgen levels decrease gradually and more slowly. This process is accelerated, usually after the age of 65–70. Decreased endocrine function of the testicles may also occur in other diseases and due to the use of certain drugs [2]. Furthermore, decreased bone mineral density (BMD) and increased risk of fractures are observed with an increase in the concentration of sex hormone binding globulin (SHBG) and a simultaneous decrease in free testosterone levels. Estrogen content should also not be discounted. Its appropriate level is crucial for achieving peak bone mass, and its deficiency is directly related to increased remodeling and bone loss in men. Patients with low estradiol and free testosterone levels are most at risk of bone loss and fractures [1]. In addition, reduced levels of dehydroepiandrosterone (DHEA) are associated with reduced bone mineral density [3].

Pine pollen (Pinus pollen) is a dry, fine, light-yellow powder. It comes mainly from the species *Pinus massoniana Lamb*, *Pinus tabuliformis Carriere*, and other plants of the same species. It is produced by male inflorescences during the pine flowering period, i.e., in May, and its presence can be observed after rainfall, as it is then found on the banks of puddles, creating a yellow coating—the so-called “sulphur rains”. Pine pollen has been used in traditional medicine in China, Japan, and Korea for hundreds of years. Modern research has revealed the health properties of pollen, proving its immunomodulatory effect on the body, its protective effect on the liver, its anticancer, antioxidant, and anti-inflammatory effects, and its reduction in the effects of aging [4,5]. Pine pollen contains numerous enzymes, microelements, vitamins, flavonoids, and phytosterols. However, the most important feature seems to be the presence of testosterone and dehydroepiandrosterone (DHEA) [6]. This lets pine pollen be classified as a bona fide source of phytoandrogens [7].

This study aimed to determine the osteotropic effect of two different doses of pine pollen administered orally on bone tissue metabolism in male rats in the conditions of development of atrophic bone changes induced by bilateral orchidectomy. The parallel aim of our research was to establish the effect of gonadectomy and different doses of pine pollen on the characteristics of calf muscles.

## 2. Materials and Methods

### 2.1. Animal Procedures

All described procedures and experimental methods were approved, before starting the experiment, by the Ethical Committee for Animal Experiments in Lublin, Poland (resolution no. 92/2019). This experiment was carried out in compliance with the ARRIVE guidelines.

This study was performed in Experimental Medicine Center (EMC) of the Medical University of Lublin Poland, which has a Certificate of Good Laboratory Practice (registration number 24/2020/DPL of 2 December 2020). Healthy, 3-month-old male Wistar rats (n = 40), originating from EMC, weighing 300 g ± 20 g (no other inclusion or exclusion criteria were used) were placed in numbered standard cages for this species, two animals in one cage. The animals also came from the Experimental Medicine Center of the Medical University of Lublin Poland. The rats were kept in optimal conditions for the species: temperature (22 °C ± 2), lighting (day/night 12/12 h), and humidity (55% ± 5). During 7 days of acclimatization and 60 days of experimental period the animals had unlimited access to standard maintenance feed Altromin Standard 1320 (Altromin, Lage, Germany) and water. The animals were also accustomed to the touch and smell of humans. The rats were taken out of the cage every day and petted for a few minutes. This activity allowed to minimize the animals’ stress. The cages were also equipped with elements that enriched the rodents’ environment (plastic houses, cardboard tubes, nesting material).

Animal selection was performed using simple randomization. The method involved assigning a number from 1 to 40 to each animal. Then, a random number was generated for each rat using the RAND function in Excel. Random numbers were sorted from lowest to highest, and then the sorted rats were assigned to groups. In the first stage, 40 male rats were randomly divided, of which n = 8 were selected for the SHO group and n = 32 for the ORX group. After a 7-day convalescence, ORX rats (n = 32) were assigned to the particular ORX groups based on the same method. During SHO and ORX surgeries, Isotek (Vet-Agro, Lublin, Poland) was used for inhalation anesthesia of the rats. The SHO procedure consisted of a gentle incision of the scrotal skin along the long axis of each testicle in such a way as not to damage the gonadal tissues, and then the wound edges were closed using 5–0 nonabsorbable sutures. During the ORX, similar incisions were made as described above, through which the testicles were removed. After surgical procedures, Melovem (Dopharma B.V., Raamsdonksveer, The Netherlands) was used to relieve pain in animals at a dose of 2 mg/kg b.w. administered subcutaneously. To prevent possible inflammation, an antibiotic in a dose of 0.5 mL (Betamox, ScanVet Sp. z o.o, Skiereszewo, Poland) was once administered directly to the wound. After postoperative convalescence, the ORX rats were randomly divided into 4 groups (n = 8): negative control treated by gavage with physiological saline (Polfa S.A., Warszawa, Poland) (ORX-PhS), positive control received exclusively testosterone in subcutaneous injection (Polfa, Poland) (ORX-TEST), and two groups treated with different doses of pine pollen applied by gavage (Asiya Life Company Ltd., Samut Sakhon, Thailand) (Table 1). The number of animals in each group was determined in accordance with EU legal regulations on the protection of animals used for scientific purposes and the recommendations of the Ethical Committee for Animal Experiments, Lublin, Poland. The correctness of the selection of the sample size was confirmed by the power analysis of the statistical test, where at α = 0.05 1 − β = 1 using Statistica version 13.3 PL software (Tibco, Palo Alto, CA, USA).

The pine pollen doses used in this experiment were established based on a query in literature databases (Medline and Scopus) [8,9,10] and on the basis of pilot studies (data not presented). The body weight of the rats, which was the basis for determining the doses of the substances used, was measured once a week. After 60 days of this experiment, the animals were euthanized by decapitation. The left tibiae were isolated and cleaned of soft tissue, while the right pelvic limbs were separated from the body at the hip joint, together with the skin and muscles. Blood was collected from a cardiac puncture, and after centrifugation, the obtained plasma was used for biochemical analyses.

### 2.2. Dual X-Ray Absorptiometry (DXA) Measurements of the Total Skeleton and Isolated Tibia

Densitometric measurements were performed by employing a Norland Excel Plus densitometer (Norland, Ford Atkinson, WI, USA) equipped with Norland Illuminatus software version 4.7.6. The bone mineral content, bone mineral density, and surface of bone tissue of the total skeleton (Ts.BMC, Ts.BMD, and Ts.Ar), as well as of isolated tibia (t.BMC, t.BMD, and t.Ar) were determined. Before each series of measurements, the device was calibrated according to the manufacturer’s recommendations [11].

### 2.3. Analysis of Isolated Tibia Using Peripheral Quantitative Computed Tomography (pQCT)

Isolated tibiae were analyzed using a pQCT Stratec XCT Research SA Plus tomograph controlled by Stratec software, version 6.20 C (Stratec Medizintechnik GmbH, Pforzheim, Germany). The densitometric and structural parameters of trabecular and cortical bone tissues were measured in the proximal metaphysis and middle of the shaft of a column, respectively, as described previously [12,13]. The detailed methodology and parameters analyzed in cortical and trabecular bone tissue are presented in Appendix A and Table A1.

### 2.4. Muscle Tissue Analysis Using Peripheral Quantitative Computed Tomography (pQCT)

Muscle tissue analysis was performed on the right pelvic limb, which was isolated from the trunk at the hip joint. The measurement site was 50% of the length of the tibia. The obtained results were the basis for further calculations, allowing for the determination of muscle cross-sectional area (mCSA), muscle density (MD), intramuscular adipose tissue area (IMAT), and subcutaneous adipose tissue (SAT) according to Frank-Wilson et al. [14].

### 2.5. Analysis of Bone Mechanics

Isolated tibiae were subjected to strength measurements (three-point bending test) via a ZwickRoell Z010 device (Zwick GmbH&Co. KG, Ulm, Germany), controlled by testXpert II software version 3.1 and equipped with a measuring head (Xforce HP series) with a range of work from 0 to 10 kN. On the day of testing, each bone was thawed at room temperature before analysis and then placed horizontally with the anterior side facing down on two transverse supports and central along its length. The distance between supports was set individually for each sample at 40% of the total bone length. The bone dimensions, i.e., length and external and internal diameters of the shaft at the point of action of the bending head, were measured using pQCT, as previously described [15]. The data were entered into the program controlling the operation of the testing machine, which allowed the use of a pipe model instead of a rod for testing. The preload for each sample was set to 10 N at a crosshead speed of 0.1 mm/s. The reference point in the tests was the loading force at a constant measuring head speed of V = 10 mm/min. For each bone, the maximal force (F_max_), the Young modulus of elasticity (E_mod_), and the force of elastic limit (F_r_) were assessed [16]. Using pQCT, the value of the axial strength–strain index (xSSI) was also calculated. This method allows for determining the predicted mechanical strength of the bone without sample destruction.

### 2.6. Biochemical Markers of Bone Metabolism

Assessment of bone metabolic parameters included measurement of serum levels of C-terminal telopeptide of type I collagen (CTX-I) (catalog no. AC-06F1—Immunodiagnostic Systems, Bolton, UK), osteocalcin (OC) (catalog no. AC-12F1—Immunodiagnostic Systems, Bolton, UK), and bone-specific alkaline phosphatase (bALP) (catalog no. E-EL-R1109—Immunodiagnostic Systems, Bolton, UK). Analyses were accomplished by utilizing the above-mentioned commercial enzyme-linked immunosorbent assays (ELISAs) and a Benchmark Plus microplate reader (Bio-Rad Laboratories Inc., Hercules, LA, USA).

### 2.7. Statistical Analysis

The values of the analyzed parameters were presented as mean values with the standard error of the mean (x ± S.E.M.). The normality of the data distribution was verified using the Shapiro–Wilk test. Multivariate comparisons were executed by applying a one-way analysis of variance (ANOVA). After finding significant statistical differences between individual groups, Tukey’s post hoc tests were undertaken to examine them more precisely. Correlation matrices of strength parameters were also determined based on the correlation analysis from the basic statistics block. The statistical significance coefficient was assumed as *p* < 0.05. Statistical analysis of the obtained results was performed using the Statistica version 13.3 PL software (Tibco, Palo Alto, CA, USA).

## 3. Results

### 3.1. Densitometric Analysis (DXA) of the Total Skeleton and Isolated Tibia

Orchidectomy significantly reduced Ts.BMC and Ts.BMD in the ORX-PhS group (*p* < 0.01 vs. SHO) (Figure 1). Significantly lower Ts.BMC and Ts.BMD values were also observed in the ORX PP50 group (*p* < 0.01).

Application of pine pollen at the dose of 150 mg/kg b.w. maintained the Ts.BMC and Ts.BMD values as in the SHO and ORX-TEST groups. Neither orchidectomy, testosterone, nor various doses of pine pollen affected the total skeletal area of rats (Ts.Ar). A similar relationship was observed in the DXA measurements of the t.BMC, t.BMD, and t.Ar of the isolated tibia (Figure 1).

### 3.2. Tomographic Analysis of Cortical Bone Tissue at Mid-Length of the Tibia

Significantly, the lowest values of Ct.BMC and Ct.vBMD of the tibia shaft were noted in the ORX-PhS and ORX-PP50 groups (*p* < 0.05) (Table 2). A higher dose of pine pollen inhibited the atrophic changes of the cortical bone tissue and allowed the maintenance of the values of the assessed parameters at the level of the SHO and ORX-TEST groups. No significant differences were found in the size of the pericortical circumference (PERI.C) of the tibia shaft. In contrast, the endocortical circumference (ENDO.C) was significantly greater in the ORX-PhS and ORX-PP50 groups compared to the SHO, ORX-TEST, and ORX-PP150 groups (*p* < 0.05) (Table 2). The consequence of changes in the size of the PERI.C and ENDO.C circumferences were changes in the thickness of the cortical bone layer (Ct.Th). Significantly, the lowest Ct.Th values were found in the ORX-PhS and ORX-PP50 male groups (*p* < 0.05) as compared to other groups of this experiment. The thickness of the cortical bone layer in the SHO, ORX-TEST, and ORX-PP150 groups did not show significant differences in values (Table 2).

### 3.3. Analysis of Trabecular Bone Tissue in the Proximal Metaphyseal Part of the Tibia Using pQCT

Separate tomographic evaluation of trabecular bone tissue shows a significant effect of gonadectomy on Tb.BMC and Tb.vBMD of the tibia in males. Elimination of hormonal influence of gonads reduced Tb.BMC by 42% (*p* < 0.00001) and Tb.vBMD by 31% in the ORX-PhS group (*p* < 0.0001), as compared with the SHO group (Table 3). Moreover, significantly lower values of these parameters were evident in the ORX-PP50 group, with differences of 30% (*p* = 0.0009) and 21% (*p* = 0.0002), respectively (vs. SHO). It should be noted, however, that the values obtained in the ORX-PP50 group were significantly higher than those in the ORX-PhS control group by 20% and 14%, respectively (*p* < 0.05). Despite the administration of testosterone and pine pollen at a dose of 150 mg/kg b.w., a decrease in the Tb.BMC and Tb.vBMD values of the proximal metaphyseal part of the tibia (vs. SHO) were seen in these groups, with no statistical confirmation of differences in mean values (Table 3).

### 3.4. Tomographic Measurements of Muscle Tissue

Orchidectomy of rats statistically significantly reduced mCSA (*p* < 0.00001) and MD (*p* < 0.00001). Furthermore, the use of a lower dose of pine pollen statistically significantly reduced mCSA (*p* = 0.0004) and MD (*p* < 0.00001) compared to the SHO group, and the obtained values of these parameters did not differ from those recorded in the ORX-PhS group (Table 4). Herein, the administration of testosterone and a higher dose of pine pollen inhibited muscle atrophy, as expressed by similar values of mCSA and MD to those in the SHO group. In addition, orchidectomy statistically significantly increased the IMAT area, which was confirmed in the ORX-PhS group (*p* = 0.0003). What is more, the lower dose of pine pollen did not limit the accumulation of adipose tissue in muscles caused by orchidectomy, and its surface was almost 12% larger than in the SHO group (*p* = 0.0008).

The use of testosterone and a higher dose of PP was also marked by a slight but not statistically confirmed increase in IMAT area compared to the SHO group. Moreover, orchidectomy increased the surface of subcutaneous adipose tissue by as much as 85% in the ORX-PhS group (*p* < 0.00001). Finally, in the other groups of rats subjected to orchidectomy, the surface of adipose tissue was statistically significantly larger as compared to the control SHO group (*p* < 0.05) (Table 4).

### 3.5. Assessment of the Mechanical Strength of the Tibia Performed Using the Three-Point Bending Test

Castration significantly reduced the mechanical strength of the tibia, as shown by statistically significantly lower values of maximum force (F_max_), elastic limit (F_r_), and Young’s modulus of elasticity (E_mod_) in the ORX–PhS control group and the group receiving pine pollen at a dose of 50 mg/kg b.w., compared to the other groups of this study (*p* < 0.05). The use of testosterone and pine pollen at a dose of 150 mg/kg b.w. maintained the values of the mechanical parameters at the same level as in the SHO group (Figure 2). A similar relationship of changes to that observed during measurements of strength parameters using a loading machine was noted in a study employing pQCT, in which the axial strength–strain index (xSSI) was determined, which allows for the prediction of mechanical properties without damaging bone tissue. Also, in this analysis, the lowest values were found in the ORX-PhS and ORX-PP50 male groups (*p* < 0.05), while the xSSI values of the tibia of males in the SHO, ORX-TEST, and ORX-PP150 groups were at a similar level (Figure 2).

### 3.6. Evaluation of Biochemical Markers of Bone Metabolism

The OC concentration measurements indicate a significant decrease in its concentration in the blood of rats in the ORX-PhS and ORX-PP50 groups (*p* < 0.05). In the SHO, ORX-TEST, and ORX-PP150 groups, the OC levels were similar. The same relationship was observed in the concentration of the bone alkaline phosphatase fraction (bALP) (Figure 3). It should be noted, however, that the bALP concentration in the ORX-PP50 group was 16% higher than in the control ORX-PhS group. Indeed, the administration of a higher dose of pine pollen resulted in a nearly 10% increase in the bALP level vs. SHO. The most intensive bone resorption, measured by the highest CTX-I concentration in the blood serum, was observed in the ORX-PhS and ORX-PP50 groups, which was statistically confirmed in relation to the other groups (*p* < 0.05). In the groups where testosterone and pine pollen were applied in a higher dose, the CTX-I concentration did not differ from the level of this indicator measured in the SHO group (Figure 3).

## 4. Discussion

Osteoporosis in men is one of the important medical problems concerning this sex. Despite the differences in the course of this disease between women and men, the medical consequences and adopted therapeutic procedures are similar. However, it should be emphasized that patients experience numerous side effects as a consequence of receiving antiosteoporotic drugs [17]. Hence, there is a growing interest in using substances of natural origin or their synthetic derivatives. The presented studies demonstrate the potential osteoprotective effectiveness of different doses of pine pollen administered to male rats in the conditions of development of atrophic changes in bone tissue conditioned by bilateral orchidectomy. Using rats as model animals is an acknowledged procedure in this type of research [18]. The changes occurring in bone tissue in rats under experimental gonadectomy conditions resemble those observed in postmenopausal women and post-andropausal men [19,20,21,22].

Eliminating the influence of steroid sex hormones in males significantly affects bone tissue metabolism, which results in a significant reduction in the mineral content of the total skeleton and isolated tibia. This outcome follows the results of other studies [23,24,25]. The osteoprotective effects of pine pollen result from the applied dose. DXA measurements indicate that the dose of 50 mg/kg b.w. is insufficient to limit the loss of minerals in the total skeleton and isolated tibia, which leads to the intensification of atrophic changes in bones manifested by a significant reduction in mineral content and density. The dose of 150 mg/kg b.w., however, effectively inhibited atrophic changes, which resulted in maintaining the Ts.BMC and t.BMC values at the level of the sham-operated control group. A similar relationship was observed for Ts.BMD and t.BMD. However, no significant effects of orchidectomy and pine pollen on the surface of the entire skeleton and tibia were found.

In contrast to DXA, the pQCT method enables the measurement of bone densitometry, separately for cortical and trabecular bone tissue, and enables the analysis of bone geometry [26,27]. Cortical bone tissue is characterized by high resistance to factors conditioning its degradation, as a result of which atrophic changes observed in conditions of experimentally induced osteopenia are insignificant. Yeh et al. [28] found no significant changes in densitometric and structural parameters of the bone diaphysis in orchidectomized rats. It should be noted, however, that the authors assessed changes occurring in the bones as early as 4 weeks after gonadectomy. This period seems to be too short in rats, but it is sufficient in male mice, as demonstrated by Moverare et al. [29]. The methodological assumptions of our studies assumed 60 days of development of osteopenic atrophic changes [30]. As a result, the decrease in Ct.vBMD and Ct.BMC was found, which follows the observations of other authors [31]. The effects of pine pollen application on cortical bone tissue were dose dependent. A higher dose proved effective in preventing the development of atrophic changes in cortical bone tissue, maintaining the degree of its mineralization at the same level as in the control SHO rats and those receiving testosterone. Geometric changes of the shaft assessed in 50% of the length of the tibia are also interesting. Orchidectomy did not significantly affect periosteal bone resorption. Thus, no significant differences were found in the size of the pericortical circumference of the tibia shaft between the individual groups of this experiment. However, an increase in the size of the endocortical circumference of the shaft was noted. This indicates the intensification of resorption processes within the marrow cavity. However, this increase was observed only in the control group of orchidectomized rats (ORX-PhS) and in the group treated with pine pollen at a dose of 50 mg/kg b.w. These observations are consistent with the results published by other authors [32]. The consequence of bone metabolism occurring under the periosteum and within the marrow cavity is a reduction in the thickness of the cortical bone layer (Ct.Th) in the ORX-PhS and ORX-PP50 groups, while the tibia Ct.Th in the ORX-PP150 group did not differ from that found in the SHO and ORX-TEST groups [33].

The results of our studies (Figure A1) and those of other authors show that trabecular bone tissue is much more sensitive to any interactions that affect its metabolism [28]. Tomographic analysis of trabecular bone tissue of the tibia in ORX-PhS group revealed a significant reduction in the degree of mineralization, as expressed by lower values of Tb.BMC (42%), Tb.vBMD (31%), and Tb.Ar (17%) in comparison with the SHO. Significantly lower mineralization (Tb.BMC and Tb.vBMD) and Tb.Ar was also noted in the ORX-PP50 group. These differences were 30%, 22%, and 7.5%, respectively. Despite the intensive resorption of the trabecular bone tissue in the ORX-PP50 group, it should be noted that this dose of pine pollen had a partial osteoprotective effect, which was expressed by significantly higher mineralization (Tb.BMC and Tb.vBMD) and Tb.Ar compared to the group of orchidectomized control males (ORX-PhS). Effective reduction of atrophic changes in trabecular bone tissue was observed in the group receiving pine pollen at a dose of 150 mg/kg b.w., even though Tb.BMC and Tb.vBMD of the examined bones showed a tendency to lower values vs. SHO.

The “mechanostat” theory, also known as “bone biomechanics”, proposed by Frost et al. in 1998 [34] suggests that osteoblasts and osteoclasts, which are cells responsible for bone modeling and remodeling, have a major impact on bone geometry. These processes are controlled by endocrine–metabolic factors and result from the impact of muscle force [35]. In this way, bones can adapt to the maximum forces generated by muscles in different parts of the body under physiological conditions by removing and adding bone tissue in different areas of the skeleton [36]. Testosterone deficiency in men is associated with a loss of skeletal muscle mass, although the mechanism is poorly understood. Wanga et al. [37] showed that muscle strength, which directly affects the quality of bone tissue, results primarily from their density and surface area. Thus, the parallel goal of our research was to determine the effect of gonadectomy and different doses of pine pollen on the features of the calf muscles. Analysis of muscle tissue was performed using pQCT and the measurement procedure proposed by Frank-Wilson et al. [14]. The obtained results indicate that following orchidectomy, the cross-sectional area (CSA) and muscle density (MD) decreased significantly in the ORX-PhS group. The results of our study indirectly confirm the observations of Hanson et al. who showed a decrease in muscle mass and strength in orchidectomized rats [38]. Following orchidectomy, the area of subcutaneous adipose tissue (SAT) and intramuscular adipose tissue (IMAT) increased, which is also confirmed by other studies [39]. Moreover, muscle weakness and increased adipose tissue content were found in the group using the lower dose of pine pollen.

The administration of pine pollen at a dose of 50 mg/kg b.w. had a noticeable protective effect on muscles, which was reflected in a smaller difference in the analyzed parameters within ORX-PhS, compared to the SHO control. Pine pollen at a dose of 150 mg/kg b.w. was most effective as protection. The density and surface area of the tested muscles were similar to the values recorded in the SHO group. It also limited the accumulation of subcutaneous and intramuscular adipose tissue.

The relationship between the quality of bone and muscle tissue was confirmed by correlation analysis, which revealed that mCSA and MD are significantly and positively correlated with both the content and mineral density of the entire skeleton and isolated tibia (Table 5). However, a negative, significant correlation was shown when comparing the DXA parameters of the skeleton and isolated tibia with IMAT and SAT.

The definition of osteoporosis emphasizes that with the development of atrophic changes in bone tissue, the susceptibility of bones to fractures increases [40]. We demonstrated that eliminating the influence of gonadal hormones reduces the mechanical strength of bones, which is reflected in the values of the assessed parameters. Similar results were presented by other authors [41]. Using a higher dose of pine pollen effectively stopped the decrease in mechanical properties, and the strength of the tibia bones was at the level of the SHO and ORX-TEST groups.

The examination of bone strength using the three-point bending test as a dynamic load, due to its damaging nature, is only possible postmortem. The in vivo method that allows for predicting mechanical strength without bone damage is peripheral quantitative computed tomography (pQCT). Using it, it is possible to determine the axial force–strain index (xSSI) and to predict the mechanical strength of the bone [42]. In our studies, the xSSI measurement was performed in the same location as the subsequent strength test in the three-point bending test. The nature of the xSSI index changes was identical in comparison with the parameters of the actual mechanical measurements of the isolated tibia. The usefulness of xSSI for in vivo predicting mechanical properties of bone supports a positive and statistically significant (*p* < 0.05) correlation between xSSI vs. E_mod_, F_max_ as well as F_r_ (Figure A2 in Appendix B).

However, our study has some important limitations. The results of endocrine tests, especially serum testosterone, DHEA and estradiol levels, were not presented in this manuscript. In our opinion these results are worth presenting in the context of a more complete analysis of the hormonal and biochemical profile, as well as oxidative stress. The content of phytoandrogens and phytosterols in pine pollen was also not analyzed. It is worth emphasizing that the composition and ranges of phytoandrogens and isoflavones are well known and described in the literature, taking into account the changes in their content in different pine species. This may limit the repeatability of the composition of pine pollen during its use. On the other hand, the randomness of pollen choosing was an important element of this experiment, because such a choice is made by consumers.

## 5. Conclusions

The results of the studies provide clear evidence of the osteoprotective effect of pine pollen, depending on the applied dose. Using a dose of 150 mg/kg b.w. in rats with developing osteopenia completely inhibited the development of atrophic changes in bones caused by the lack of the influence of gonadal hormones. This relationship was shown both in the cortical and trabecular bone tissue. What is more, the administration of a dose of 50 mg/kg b.w. to orchidectomized rats was also marked by a protective effect on bones. However, it was less pronounced and concerned only trabecular bone tissue. A higher dose of pine pollen inhibited the catabolic effect of orchidectomy on the calf muscles, which was expressed by maintaining their density and surface area at the level of the sham-operated group and the group receiving testosterone. Its limiting effect on the accumulation of intramuscular and subcutaneous adipose tissue was also noted. It is worth emphasizing the osteoprotective effectiveness of pine pollen, especially when administered in larger doses, which proves the possibility of its use in the prevention of the development of osteoporosis in men and males of other animal species.

## Figures and Tables

**Figure 1 nutrients-17-02110-f001:**
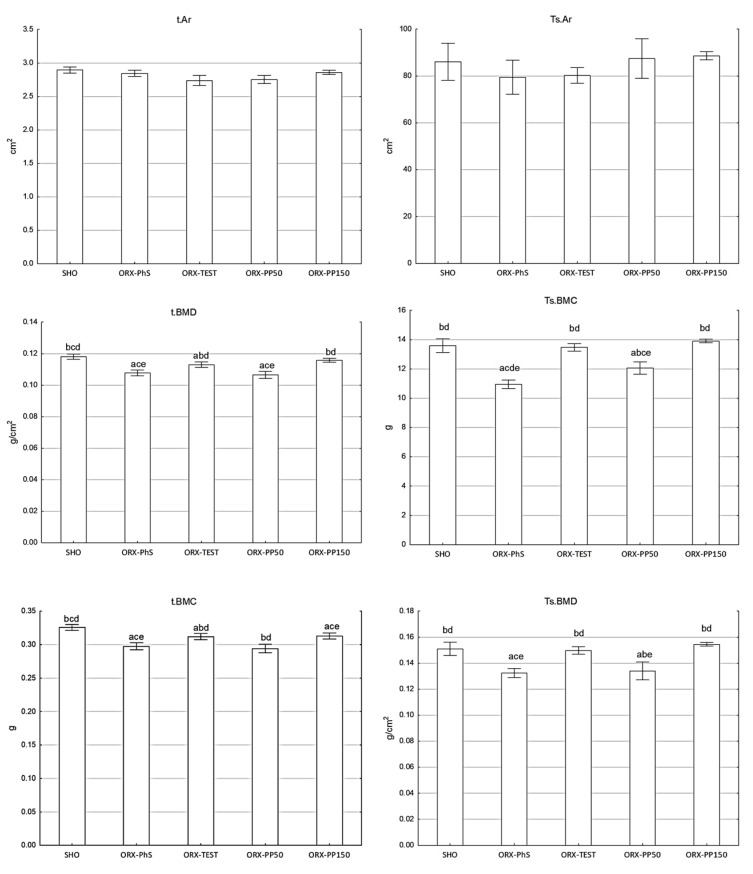
DXA parameters (area, BMC, and BMD) of the total skeleton (Ts) and isolated tibia (t). Explanation: a vs. SHO; b vs. ORX-PhS; c vs. ORX-TEST; d vs. ORX-PP50; and e vs. ORX-PP150.

**Figure 2 nutrients-17-02110-f002:**
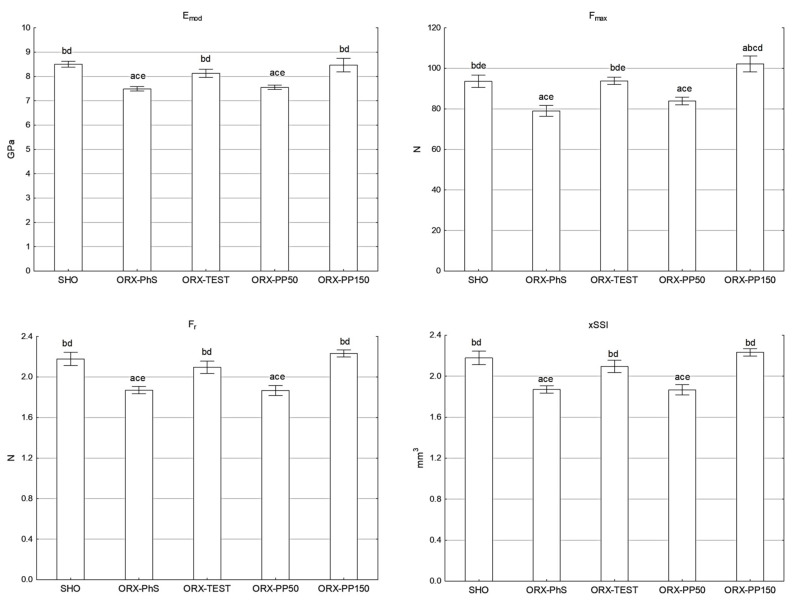
Maximal force (F_max_), the Young’s modulus of elasticity (E_mod_), the force of elastic limit (F_r_), and pQCT (xSSI) of tibia shaft. Explanation: a vs. SHO; b vs. ORX-PhS; c vs. ORX-TEST; d vs. ORX-PP50; and e vs. ORX-PP150.

**Figure 3 nutrients-17-02110-f003:**
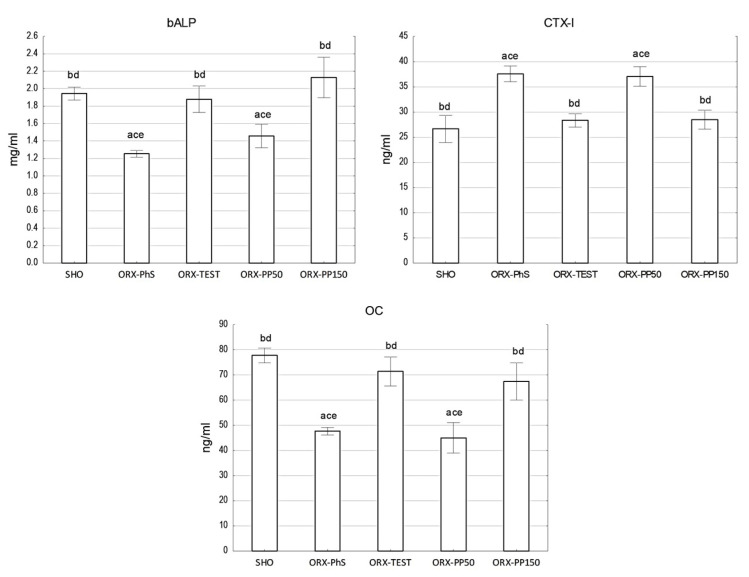
Biochemical parameters of bone tissue metabolism. Explanation: a vs. SHO; b vs. ORX-PhS; c vs. ORX-TEST; d vs. ORX-PP50; and e vs. ORX-PP150.

**Table 1 nutrients-17-02110-t001:** Description of experimental groups.

Group	Surgery	Experimental Treatment	Dose and Way of Application
SHO (n = 8)	Sham operation	Physiological saline	0.5 mL/100 g b.w./24 h *p.o.*
ORX-PhS (n = 8)	Orchidectomy	Physiological saline	0.5 mL/100 g b.w./24 h *p.o.*
ORX-TEST (n = 8)	Orchidectomy	Testosterone	7 mg/ kg b.w./7 days *s.c.*
ORX-PP50 (n = 8)	Orchidectomy	Pine pollen	50 mg/kg b.w./24 h in a PhS suspension, administered in volume 0.5 mL/100 g b.w. *p.o.*
ORX-PP150 (n = 8)	Orchidectomy	Pine pollen	150 mg/kg b.w./24 h in a PhS suspension, administered in volume 0.5 mL/100 g b.w. *p.o.*

**Table 2 nutrients-17-02110-t002:** pQCT analysis of the total cross-section of the bone shaft and cortical bone tissue determined at mid-length (50%) of the tibia.

Parameter	SHO	ORX-PhS	ORX-TEST	ORX-PP50	ORX-PP150
Tot. BMC—(mg/mm)	6.41 ± 0.07 bd	5.97 ± 0.03 ace	6.29 ± 0.04 bd	5.82 ± 0.1 ace	6.35 ± 0.08 bd
Tot. vBMD (mg/mm^3^)	1054.9 ± 12.0 bd	996.5 ± 20.0 ace	1047.7 ± 14.8 bd	1009.9 ± 12.0 ace	1071.9 ± 5.6 bd
Tot.Ar (mm^2^)	6.12 ± 0.12	5.96 ± 0.08	5.91 ± 0.09	6.06 ± 0.05	6.17 ± 0.14
Ct. BMC (mg/mm)	5.95 ± 0.11 bd	5.42 ± 0.07 ace	5.82 ± 0.1 bd	5.56 ± 0.07 ace	5.92 ±0.06 bd
Ct.vBMD (mg/mm^3^)	1373.5 ± 13.4 b	1329.0 ± 14.7 ace	1372.2 ± 9.2 b	1359.4 ± 12.4 e	1393.7 ± 5.1 bd
Ct. Ar (mm^2^)	4.78 ± 0.18	4.38 ± 0.20	4.64 ± 0.15	4.30 ± 0.19	4.71 ± 0.17
Ct. Th (mm)	0.72 ± 0.02 bde	0.64 ± 0.01 ace	0.69 ± 0.01 bd	0.62 ± 0.01 ace	0.69 ± 0.01 abd
Peri.C (mm)	8.83 ± 0.08	8.73 ± 0.15	8.72 ± 0.15	8.64 ± 0.09	8.87 ± 0.13
Endo.C (mm)	4.22 ± 0.16 bd	4.56 ± 0.07 ace	4.22 ± 0.08 bd	4.54 ± 0.08 ac	4.27 ± 0.04 b

Explanation: a vs. SHO; b vs. ORX-PhS; c vs. ORX-TEST; d vs. ORX-PP50; and e vs. ORX-PP150.

**Table 3 nutrients-17-02110-t003:** pQCT analysis of the total cross-section and trabecular bone tissue in proximal tibia metaphysis.

Parameter	SHO	ORX-PhS	ORX-TEST	ORX-PP50	ORX-PP150
Tot. BMC—(mg/mm)	12.0 ± 0.4 bcd	10.1 ± 0.2 ace	11.2 ± 0.2 bd	10.7 ± 0.1 ae	11.7 ± 0.1 bd
Tot. vBMD (mg/mm^3^)	613.2 ± 15.0 bd	549.5 ± 9.0 ace	588.9 ± 9.7 bd	554.1 ± 11.1 ace	603.1 ± 4.8 bd
Tot.Ar (mm^2^)	19.5 ± 0.5 bd	15.7 ± 0.4 ace	18.2 ± 0.7 bd	17.0 ± 0.4 ae	20.0 ± 0.2 bcd
Tb. BMC (mg/mm)	2.3 ± 0.2 bd	1.4 ± 0.1 acde	2.1 ± 0.2 bd	1.6 ± 0.1 abce	2.2 ± 0.1 bd
Tb.vBMD (mg/mm^3^)	264.6 ± 18.6 bd	182.2 ± 5.0 acde	236.8 ± 12.5 b	207.7 ± 2.0 ab	232.9 ± 3.7 b
Tb.Ar (mm^2^)	8.8 ± 0.2 bd	7.3 ± 0.2 acde	8.6 ± 0.3 b	8.1 ± 0.2 abe	9.0 ± 0.1 bd

Explanation: a vs. SHO; b vs. ORX-PhS; c vs. ORX-TEST; d vs. ORX-PP50; and e vs. ORX-PP150.

**Table 4 nutrients-17-02110-t004:** pQCT analysis of muscle cross-section performed in 50% of the length of the lower leg.

Parameter	SHO	ORX-PhS	ORX-TEST	ORX-PP50	ORX-PP150
mCSA (cm^2^)	2.0 ± 0.03 b	1.7 ± 0.01 ace	1.9 ± 0.03 b	1.8 ± 0.02 ae	1.90 ± 0.02 bd
MD (mg/cm^3^)	84.4 ± 0.2 bd	82.4 ± 0.2 ace	83.8 ± 0.1 bde	82.5 ± 0.2 ace	84.3 ± 0.2 bcd
IMAT (cm^2^)	0.384 ± 0.008 bde	0.442 ± 0.004 ace	0.406 ± 0.007 b	0.429 ± 0.003 a	0.411 ± 0.003 ab
SAT (cm^2^)	0.058 ± 0.003 bcde	0.087 ± 0.004 acde	0.071 ± 0.003 ab	0.077 ± 0.002 abe	0.066 ± 0.002 abd

Explanation: a vs. SHO; b vs. ORX-PhS; c vs. ORX-TEST; d vs. ORX-PP50; and e vs. ORX-PP150.

**Table 5 nutrients-17-02110-t005:** Correlation table of densitometric parameters of the whole skeleton and isolated tibia in relation to the parameters of tomographic muscle assessment.

Parameter	MD (mg/cm^3^)	mCSA (cm^2^)	IMAT (cm^2^)	SAT (cm^2^)
Ts.BMD (g/cm^2^)	0.4914 *p* = 0.002	0.5037 *p* = 0.001	−0.5079 *p* = 0.001	−0.3697 *p* = 0.024
Ts.BMC (g)	0.6167 *p* = 0.000	0.6495 *p* = 0.000	−0.6306 *p* = 0.000	−0.5409 *p* = 0.001
t.BMD (g/cm^2^)	0.5087 *p* = 0.001	0.5527 *p* = 0.000	−0.5261 *p* = 0.001	−0.4151 *p* = 0.011
t.BMC (g)	0.4587 *p* = 0.004	0.4817 *p* = 0.003	−0.5121 *p* = 0.001	−0.4120 *p* = 0.011

Explanations: The numerical values in the table present the correlation coefficients r and the value of the significance level *p*. All values of the correlation coefficients in the table are statistically significant at *p* < 0.05.

## Data Availability

Data available upon reasonable request. The data are not publicly available due to this research being part of a larger research project.

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
