# Peer review of "Osteoprotective Effect of Pine Pollen in Orchidectomized Rats"

_nutrients, 2025, doi:10.3390/nu17132110_

Round 1

Reviewer 1 Report

Comments and Suggestions for Authors

Major concerns
1. Chemically standardise and quantify the pollen batch (e.g., GC-MS for testosterone/DHEA, LC-MS for flavonoids) and report concentrations. The authors rely on literature to claim pine-pollen contains testosterone/DHEA but never analyse the material they gavaged. 
2. Clarify dosing frequency for pine pollen and add an oral-vehicle + s.c. sham control to disentangle route and stress. Table 1 gives the gavage volume but omits frequency (“/24 h”), unlike the saline control. Testosterone is injected s.c. once per week, whereas pollen and saline are given by oral gavage; no sham injection control is provided.
3. Provide an a-priori power calculation (effect size, α = 0.05, 1-β ≥ 0.8); consider increasing n or reducing endpoints.
4. Add serum androgen and pollen-derived phyto-sterol measurements to substantiate the mechanistic narrative. Serum testosterone/DHEA are not measured, so pollen’s “phytoandrogen” mechanism remains speculative.
5. Re-analyse data with SD and family-wise error adjustment; highlight effect sizes (Cohen’s d) beside p-values. Dozens of DXA, pQCT, biomechanical and serum markers are analysed with one-way ANOVA + Tukey; no family-wise correction (e.g., Holm–Bonferroni) is applied. It elevates risk of type-I error.
6. Expand Methods for mechanical testing (support span, gauge length, preload) and specify all pQCT scanning parameters.
7. Tighten language in Discussion—differentiate statistical from biological significance and temper claims of human applicability.
8. Correct typographical, unit and abbreviation inconsistencies and supply complete figure axes, legends and supplementary links. Ambiguous/incorrect group codes. “ORC-PP50” appears instead of ORX-PP50 (line 319), and “shame-operated” is listed in abbreviations. Commas used as decimal separators in tables (e.g., 613,2 mg/mm³) alongside points, and mixed mg · mm vs mg/mm units. Figures 1–3 use colour/shape codes but lack explicit axis units; group labels overlay data points in Figure 2.

Minor concerns
1. Only ORX rats receive a peri-wound Betamox dose; SHO rats apparently do not. β-lactams can influence bone modelling.
2. Three-point bending span length and preload are not reported; strength estimates are therefore non-reproducible.
3. Authors call a 3 % fall in cortical vBMD “statistically significant” but still claim strong osteoprotection later in Discussion. Effect size is small and may be within machine precision. Biological significance overstated.
4. Conclusion claims pollen “demonstrates the possibility of its use in the prevention of osteoporosis in men” despite single-sex rat data.

Author Response

Lublin, Poland. 11 June 2025

Editorial Office - Nutrients

Firstly, the authors would like to express their gratitude to the reviewer for efforts in preparing the revision of our manuscript. In response, we would like to present our rebuttal to the opinions expressed by the Reviewer.

Major concerns

Ad 1.

This experiment aimed to find out the effects of randomly selected pine pollen on the skeletal system of male rats under specific biological conditions. For research purposes (information supplemented in the manuscript) we used the supplement from Asiya Life Company Limited Thailand. It is worth emphasising that it holds the USDA Organic certificate, issued by the United States Department of Agriculture (USDA), for products that meet specific standards of organic farming and production. This certificate is one of the most restrictive and comprehensive certification systems, both for food products and cosmetics.         

The composition and concentration ranges of phytoandrogens and isoflavones are well known and described in the literature, taking into account the differences in their content in pine pollen from different species of pines. It is believed that the composition of pine pollen may also be significantly influenced by the conditions in which the trees grow. As mentioned earlier, commercially available pine pollen with guaranteed quality was used in the study. The randomness of the pollen selection was an important element of this experiment because such a choice is made by consumers. Therefore, the authors did not consider it justified to analyze the pine pollen composition of the batch used in the experiment, because it is not of interest to the average consumer. It should be emphasized that pine pollen should be taken for a long time. Such a suggestion is described in Shen-nung Pen-tsao Ching (Chinese book on agriculture and medicinal plants). This makes it impossible to maintain the repeatability of the pine pollen composition during the period of its use. The authors thank the reviewer for taking up this topic and inform that Material and methods has been supplemented with the source of pine pollen.

Ad 2.

The authors assumed that a weekly injection of such a small volume of testosterone, performed with an insulin syringe, is not a stressful factor for the animals and does not require the creation of a new group in the experiment. Additionally, the authors also took into account the 3R rule and the limitation of the number of animals used in the experiment. Information on the frequency of pine pollen administration has been supplemented in Table 1.

Ad 3.

The power test analysis (table below) was performed in Statistica, Tibco, USA (in Polish version) assuming α=0.05, which is according to our calculations 1-β =1.

When determining the number of animals for the experiment, the authors also took into account the 3R principle and legal regulations regarding the use of animals in scientific research (Directive 2010/63/EU of the European Parliament and of the Council of 22 September 2010 and the recommendations of the Ethical Committee for Animal Experiments in Lublin, Poland.)

Ad 4.

We would like to note that the authors of the paper do not suggest that the effect of pine pollen results from the influence of phytoandrogens contained in it, but only as a whole. Pine pollen is a mixture of numerous biologically active compounds, among which are also androgens. However, the authors would like to emphasize that this experiment aimed to assess the effect of pine pollen as a whole on the metabolism of bone tissue of orchidectomized rats, which is included in the aim of the manuscript as well as in the discussion and summary of the research. In this paper, the concentrations of androgens and phytosterols are not presented because such a presentation would be narrow, incomplete and, paradoxically, the article would lose its informativeness. Of course, such analyses were carried out and constitute an interesting study defining the effect of pine pollen on the widely studied hormonal and biochemical profile in rats and will be published soon.

We particularly appreciate the reviewer’s words: “so pollen’s “phytoandrogen” mechanism remains speculative”. We believe that every scientific study is speculative to some extent, encouraging discussion and undoubtedly providing a basis for continuing experiments. This is the main mechanism driving the development of science.

Ad 5.

The authors do not share the reviewer's view and are convinced that the statistical analyses used are correct and adequate in studies of this type. Tukey's test is highly conservative and allows for determining a truly significant difference. The use of Tukey's test undoubtedly allows for avoiding a type I error. Additionally, it controls FWER. The Bonferroni test (aka Holm-Bonferroni) is more conservative. It may seem that increasing the conservativeness is appropriate and desirable, which, as the authors believe, was the reviewer's reason for suggesting the use of this test. It should be remembered, however, that reducing the α level after applying the Bonferroni test is inextricably linked to an increased risk of making a type II error and, consequently, overlooking existing differences or relationships.

Ad 6.

We would like to thank the reviewer for pointing out this understatement. The strength testing methodology has been supplemented in the Material and Methods section.

A detailed description of the methodology of the pQCT measurements can be found in Appendix 1, while the parameters describing the cortical and trabecular bone tissue can be found in Table S1.

Ad 7. and Ad 8.

Thank you very much for this suggestion. The necessary changes have been made in the Results and  Discussion.

Minor comments

Ad 1.

In the experiment, Betamox was administered once. This information was supplemented in the Material and Methods section. To the authors’ knowledge, there are no reports of the resorptive effect of β-lactams on bone tissue after a single administration. Moreover, antibiotics based on amoxicillin, in combination with clavulanic, has promising activity against pathogens that cause bone infections. We agree, however, that long-term use, particularly at high doses, may negatively impact bone formation and potentially increase bone resorption.

Ad 2.

Thank you very much for your fair and accurate comment. Appropriate changes have been made in the text of the manuscript.

Ad 3.

We agree with the reviewer's opinion that such a low percentage difference may be questionable. The authors have rephrased this sentence.

The 3% reduction in Ct.vBMD is indeed seemingly small. However, the authors point out that in the Discussion on page 11 of the manuscript, at the beginning of the second paragraph, we emphasize the special resistance of cortical bone tissue to impacts that may affect its metabolism. In this context, the reduction in Ct.vBMD value is statistically significant. The obtained results indicate that a higher dose of pine pollen inhibits atrophic changes in cortical bone tissue, maintaining their values ​​at the level of the SHO group. However, we emphasize that the text of the manuscript does not contain the phrase "strong osteoprotection". We only note the osteoprotective effect of a higher dose of pine pollen on bone metabolism in general, and that a higher dose affects cortical bone tissue. We would also like to emphasize that the CV of pQCT analysis (XCT Research SA Plus) using rat femur and tibia from 10 measurements in our laboratory never exceeded 1.08%. This is due to our restrictive pQCT calibration procedure using dedicated phantoms. In accordance with our laboratory's quality procedures, CV coefficient assessment is performed before each series of sample measurements, and tomographic analyses are always performed by the same person.

Ad 4.

The authors thank the reviewer for his attention. Due to the fact that the authors conducted the studies on males, the statement "possibility of its use in the prevention of the development of osteoporosis in men and males of other animal species" is justified. At the same time, let us pay attention to the first paragraph of the Discussion, in which we state, citing important references, that changes in the skeletal system of rats subjected to gonadectomy are similar to those observed in humans. It should also be emphasized that in preclinical studies rats are treated as one of the model species.

Reviewer 2 Report

Comments and Suggestions for Authors

Please, see the file attached.

Author Response

Lublin, Poland. 11 June 2025

Editorial Office - Nutrients

Firstly, the authors would like to express their gratitude to the reviewer for efforts in preparing the revision of our manuscript. In response, we would like to present our rebuttal to the opinions expressed by the Reviewer.

Reviewer 2

Ad 1.

Thank you for your attention, but the authors believe that the terms osteoptropic and osteoprotective should not be considered synonymous, e.g. compound X may be osteotropic, but does not have to show osteoprotective properties. Pine pollen shows osteotropic effects and at the same time, depending on the dose used, is osteoprotective.

Ad 2.

The information has been completed.

Ad 3.

The missing information has been completed.

Ad 4.

The age of the rats has been introduced to the Material and methods

Ad 5.

The authors are very grateful for this comment. The correction of the significance record has been made. Other records were also checked for correctness.

Ad 6.

Pine pollen is a mixture of numerous biologically active compounds. In the Introduction to the manuscript, the authors listed the previously known mechanisms of action of pine pollen (immunomodulatory, antioxidant, anti-inflammatory and anticancer). The presented studies aimed to confirm the hypothesis of whether pine pollen has any effect on bone tissue metabolism in the conditions of development of atrophic changes. However, determining the actual mechanism of action of pine pollen as a whole, as well as its components, and determining the most effective dose depending on the degree of advancement of bone tissue atrophy is important and necessary, but requires further research aimed at understanding this problem, taking into account in vivo and in vitro experiments.

Round 2

Reviewer 1 Report

Comments and Suggestions for Authors

Authors addressed all issues raised by reviewer successfully. I don't have further comment on this article.